# CONCAP: Seeing Beyond English with Concepts Retrieval-Augmented Captioning

**George Ibrahim[1]**   **Rita Ramos[2]**   **Yova Kementchedjhieva[1]**

[1]Department of Natural Language Processing, MBZUAI
[2]INESC-ID, Instituto Superior Técnico, University of Lisbon
{george.ibrahim, yova.kementchedjhieva}@mbzuai.ac.ae

## Abstract

Multilingual vision-language models have made significant strides in image captioning, yet they still lag behind their English counterparts due to limited multilingual training data and costly large-scale model parameterization. Retrieval-augmented generation (RAG) offers a promising alternative by conditioning caption generation on retrieved examples in the target language, reducing the need for extensive multilingual training. However, multilingual RAG captioning models often depend on retrieved captions translated from English, which can introduce mismatches and linguistic biases relative to the source language. We introduce CONCAP, a multilingual image captioning model that integrates retrieved captions with *image-specific concepts*, enhancing the contextualization of the input image and grounding the captioning process across different languages. Experiments on the XM3600 dataset indicate that CONCAP enables strong performance on low- and mid-resource languages, with highly reduced data requirements. Our findings highlight the effectiveness of concept-aware retrieval augmentation in bridging multilingual performance gaps.

## 1 Introduction

Vision Language Models (VLMs) have achieved increasingly strong performance in image captioning, driven by the scaling of model size and the use of extensive training data (Liu et al., 2023; 2024; Tschannen et al., 2025). Although VLMs demonstrate strong performance in image captioning, most remain English-centric, lacking the ability to generate high-quality captions for other languages. Growing efforts aim to build high-quality multilingual datasets for training and evaluation and develop multilingual VLMs on these (Geigle et al., 2024; Yue et al., 2025). However, even with massive, costly multilingual training and high parameterization to accommodate multiple languages, recent models continue to exhibit substantial performance gaps between English and other languages (Yue et al., 2025).

To alleviate the need for expensive multilingual training, recent work has explored retrieval-augmented generation (RAG) as a promising direction (Ramos et al., 2023a; 2024). In this approach, models are conditioned not just on the image, but also on a set of retrieved captions in the target language. By leveraging these retrieved examples, the model requires far less multilingual training data, as it benefits from demonstrations of how to generate captions across different languages. However, a key limitation of multilingual retrieval-augmented captioning models is that retrieved captions often come from translated datasets (Thapliyal et al., 2022) and datasets sourced from Western-centric image repositories (Chen et al., 2015), which may not fully capture the nuances of the target language and culture. Furthermore, the retrieved captions come from images *similar* to the input image, but almost certainly differ from it in some aspects, which inherently introduces noise through the retrieval augmentation.

In this work, we introduce CONCAP, a multilingual image captioning model that integrates retrieved captions and *image-specific concepts*. CONCAP builds upon the mBLIP architecture (Geigle et al., 2024), augmenting generation with retrieved information from captions of sim-

ilar images and concepts relevant to the input image. This enriches the captioning process with more informative and precise context and improves the quality of generated captions. Finetuning CONCAP on just 0.6M multilingual image-caption pairs yields considerably better results on low- and mid-resource languages in the XM3600 multilingual captioning benchmark (Thapliyal et al., 2022), compared to the original mBLIP model trained on 4M samples. Despite its lightweight and data-efficient training setup, CONCAP achieves stronger multilingual capabilities on average, even compared to Pangea, a state-of-the-art multilingual VLM with 7B parameters trained on 6M data points (Yue et al., 2025). Through ablation studies of the caption and concept retrieval augmentations, we show that each component makes a valuable contribution, and the two complement each other.

## 2 Related work

### 2.1 Image Captioning

Image captioning has seen significant progress in recent years, driven by large-scale pretraining and powerful vision-and-language models (VLMs) that leverage off-the-shelf unimodal components such as CLIP encoders and language decoders (Li et al., 2023; Alayrac et al., 2022; Mokady et al., 2021; Panagopoulou et al., 2023). Very recently, captioning performance has been further enhanced by the rise of large language models (LLMs) (OpenAI, 2024; Touvron et al., 2023; Taori et al., 2023; Chiang et al., 2023) and visual instruction-tuned data (Liu et al., 2023; Laurençon et al., 2024). These models generally follow similar architectural trends. The unimodal vision and language components, typically kept frozen, are connected either through a cross-attention mechanism (e.g., Flamingo; (Alayrac et al., 2022)), trainable Q-former networks (e.g., BLIP2; (Li et al., 2023)), or a fully autoregressive architecture, where the output of the vision encoder is directly mapped to a sequence of text embeddings (e.g., LLaVA; (Liu et al., 2023)).

Despite progress in captioning quality, most models remain English-centric, while multilingual image captioning has been largely overlooked. Notable multilingual efforts include Thapliyal et al. (2022) that created XM3600, an established human-annotated captioning benchmark in 36 languages. Thapliyal et al. (2022) also proposed COCO-35 and CC3M-35L, translated versions of the standard COCO (Chen et al., 2015) and Conceptual Captions (Sharma et al., 2018) datasets in 35 languages, aimed at training multilingual captioning models. Using these datasets, they introduced BB+CC, the first large-scale multilingual image captioning model trained from scratch. Following this, mBLIP, based on BLIP2 (Li et al., 2023), introduced a more efficient multilingual vision and language model, employing the CLIP encoder (Radford et al., 2021) and multilingual LLMs. Despite having orders of magnitude fewer parameters, mBLIP is also trained on a mixture of massive machine-translated multilingual data. Very recently, Yue et al. (2025) developed PangeaIns, a culturally-aware instruction dataset with 6M samples across 39 languages, and Pangea, a multilingual multimodal large-scale model designed to bridge language and cultural gaps. However, despite extensive multilingual training, these models still demonstrate a bias toward English captioning, with substantially lower performance for other languages. In contrast, we propose a method that effectively bridges the gap between languages and bypasses the need for massive training data by leveraging retrieval.

### 2.2 Retrieval-augmented Image Captioning

Retrieval-augmented generation (RAG) methods provide additional context to the model by incorporating information retrieved from an external datastore (Lewis et al., 2020). RAG models have demonstrated improved performance not only across various NLP tasks (Fan et al., 2024; Huang & Huang, 2024) but also in vision-and-language tasks (Yasunaga et al., 2022), such as image captioning Li et al. (2024b). Closely related to our work, Ramos et al. (2023b) proposes SmallCap, showing that augmenting an image captioning model with retrieved captions not only improves captioning performance but also reduces the number of trainable parameters and facilitates adaptation to out-of-domain settings. In contrast, Li et al. (2024a) introduces the EVCap model, which incorporates retrieved concepts (e.g., object

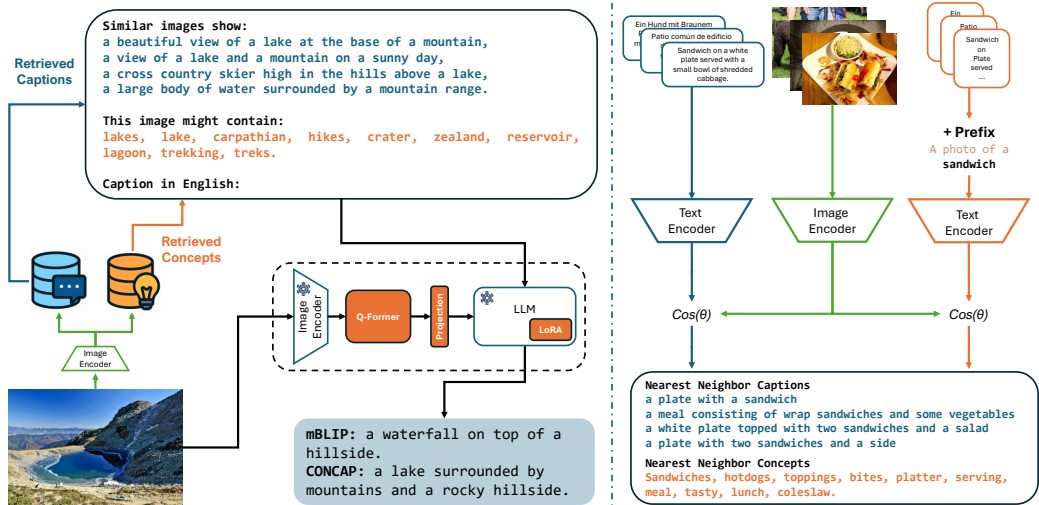

Figure 1: **(Left)** CONCAP architecture combining visual features with retrieved captions and concepts via Q-Former and LLM. **(Right)** Multilingual image-text retrieval using cosine similarity with prefixed concept prompts and cross-lingual caption mapping.

names) instead of full captions to avoid redundancy and mitigate misleading information in the retrieved text. Zeng et al. (2024) also explores augmenting image captioning with key concepts related to the image, but in a zero-shot setup.

While RAG has gained traction in image captioning, its use in multilingual image captioning has been limited, with only two models so far. Ramos et al. (2023a) explored retrieval-augmented generation in a training-free setup using in-context learning. Ramos et al. (2024) introduced PAELLA, a supervised multilingual retrieval-augmented image captioning model that is not only parameter-efficient, like SmallCap, but also data-efficient through the incorporation of retrieval augmentation. It is important to note that both of these models rely on the standard approach of using retrieved captions. However, concept-based augmentation, as seen in (Li et al., 2024a; Zeng et al., 2024), has yet to be explored in a multilingual context. Moreover, unlike previous works, we explore the integration of both retrieved captions and concept retrieval to enhance multilingual caption generation.

# 3   Proposed Approach

CONCAP is an efficient multilingual retrieval-augmented image captioning model that combines retrieved captions and retrieved concepts to generate more accurate and contextually grounded captions across languages. An overview of the approach is shown in Figure 1.

## 3.1   Architecture

The method proposed here is applicable to any vision-language modeling architecture, as it only modifies an input-specific prompt passed to the language decoder. In our experiments, we concretely adopt the mBLIP architecture (Geigle et al., 2024), one of the established state-of-the-art models for multilingual image captioning, while being parameter-efficient.

## 3.2   Caption Retrieval

CONCAP builds on standard retrieval strategies (Ramos et al., 2023b; 2024), which use vision-language representation models (Radford et al., 2021; Zhai et al., 2023) to align images and text in a shared representation space. Given a corpus of image captions in a specific language, the text encoder of a CLIP-style model is used to pre-compute and store the caption embeddings in a datastore. This indexed datastore is then queried using the

image encoder of the same model, with cosine similarity computed to find the most relevant captions based on the image embeddings.

CONCAP enhances caption generation by incorporating the top-$n$ most similar captions to the input image, guiding the language decoder towards generating captions in the target language with syntactic and semantic relevance to the intended prediction. However, we note that these retrieved captions come from similar images and may not fully capture the content of the input image, potentially introducing noise or mismatches in terms. For example, while a caption might correctly describe a bus in the street, it could mistakenly specify a "red bus," which may not be accurate for the input image.

### 3.3 Concept Retrieval

Concept retrieval follows a similar process to caption retrieval, but instead of full sentences, it concerns individual lexical items. To contextualize them and enhance their semantic representation, each concept is wrapped in a short, language-specific template (e.g., "a photo of a *dog*"). We used GPT-4o to generate the templates (OpenAI, 2024), then verified them using Google Translate. This latter step was particularly important for morphologically rich and syntactically distinct languages, where direct translations may not always yield fluent or semantically accurate prompts. A complete list of language-specific templates is provided in Figure 9. While concepts are wrapped in a template for retrieval, they are inserted into the final prompt in raw form (e.g., "*dog*").

Concept retrieval is performed analogously to caption retrieval, taking the top-$m$ most relevant concepts per image. The intuition here is that augmenting the captioning process with highly relevant retrieved concepts will counteract noise from the retrieved captions and fill in possible gaps in the concept coverage of retrieved captions.

### 3.4 Prompt Format

The final prompt for the language decoder of CONCAP combines both retrieved captions and concepts to provide informative and semantically relevant context for caption generation. The prompt is organized into three segments:

1. **Similar images show:** `caption_1, caption_2, ..., caption_n.`
2. **This image might contain:** `concept_1, concept_2, ..., concept_m.`
3. **Caption in {lang}:**

The first segment presents $n$ captions retrieved from visually similar images, providing sentence-level context. The second segment includes $m$ concepts retrieved from language-specific wordlists. Together, these components guide the model toward producing accurate and semantically grounded captions in the target language. The language of the prompt is fixed to English, and the full name of the target language is used (e.g., "Caption in Chinese".)

During training, the language decoder receives the full prompt as input and generates a caption conditioned on the image features $V$ and the task prompt $X$. The model is optimized using teacher forcing, minimizing the cross-entropy loss over the reference caption tokens.

## 4 Experiments

### 4.1 Experimental Setup

**Base Model**   CONCAP adopts the mBLIP architecture as well as its initialization strategy, i.e., *we carry out multilingual vision-language training from scratch*. The vision encoder, the Q-Former, and the projection layer are initialized from BLIP-2 (Li et al., 2023), while the language decoder is initialized from mT0-XL (Muennighoff et al., 2023). Following Geigle et al. (2024), we freeze the vision encoder and language decoder, and inject LoRA layers in the language decoder (Hu et al., 2021). Consequently, only the Q-Former, the projection layer, and LoRA layers are updated during training. This setup maintains training efficiency with approximately 111 million trainable parameters.

**Training and Evaluation Data**    For training, we use the COCO-35L dataset (Thapliyal et al., 2022), a multilingual extension of the English COCO dataset (Chen et al., 2015), where each English caption is translated into 34 additional languages using Google Translate. The dataset is based on the Karpathy split, which includes 113,287 training images, each paired with five captions, resulting in a total of 19.8M image-caption pairs across 35 languages. Following PAELLA (Ramos et al., 2024), we subsample the training split down to 566K image-caption pairs, ensuring equal representation across all languages.

We perform evaluation on the XM3600 dataset (Thapliyal et al., 2022), a human-annotated multilingual benchmark featuring captions in 36 languages. The dataset consists of 3,600 distinct images sourced from countries where the target languages are spoken, yielding a total of 261,375 captions. On average, each image is annotated with two captions.

The Pangea project (Yue et al., 2025) introduced XM100, a subset of the XM3600 benchmark, designed to enable faster and more focused multilingual evaluation while preserving diversity across languages and visual content. It consists of 100 selected images from the original set, each paired with its captions in all 36 languages.

**Caption Retrieval**    Caption retrieval is implemented differently during training and evaluation to reflect the characteristics of each dataset. COCO-35L, used for training, contains images primarily sourced from English-speaking contexts, while XM3600, used for evaluation, includes images from 36 geographically diverse countries. In both settings, we retrieve the top-4 captions per image, following prior work (Ramos et al., 2023b).

For training on COCO-35L, we adopt an English-pivot retrieval strategy following prior work (Ramos et al., 2023b; 2024). We use CLIP (Radford et al., 2021) to retrieve captions from a datastore populated with the English portion of COCO. Retrieved captions are then mapped into the target language using shared caption IDs. In §4.5, we also explore language-specific retrieval during training, which may be necessary in scenarios where English-parallel captions are unavailable.

For evaluation, retrieval is performed using the mSigLIP model (Zhai et al., 2023) directly in the target language. Since XM3600 contains geographically and culturally diverse images and captions, we expect a multilingual retriever to work better.

**Concept Retrieval**    We construct a list of concepts for each language by tokenizing the COCO-35L training captions and extracting all unique tokens. The resulting lists contain tokens that are not necessarily semantically meaningful (e.g., subword tokens) or useful (e.g., stopwords). However, curating a clean, high-quality concept list for each language is a non-trivial task, especially for low-resource languages where reliable tools for filtering or validation are limited or unavailable. Rather than relying on pre-filtering, we let the retrieval process handle relevance, trusting that the learned similarity space will naturally favor semantically meaningful and visually grounded tokens.

We maintain separate datastores per language and perform retrieval using mSigLIP. Following hyperparameter tuning over $m = 4, 10, 20$, we find that $m = 10$ yields best performance on the COCO-35L development set after 5 training epochs (see Table 5 in Appendix A.3), with results for $m = 20$ being very close.

In §5 we explore strategies for enrichment of the concept lists, and in §4.5 we compare different multilingual vision-language representation models. For efficient large-scale retrieval, we use the FAISS library (Douze et al., 2024), which performs fast nearest neighbor search over dense embeddings.

**Tokenization**    When extracting wordlists and running evaluations, we apply language-specific tokenization strategies for languages that lack explicit word boundaries, such as Chinese, Japanese, Thai, and Hindi. These languages require specialized tokenizers to accurately segment text into meaningful units. For Chinese, we use the Jieba library (Junyi, 2012); for Thai, the PyThaiNLP library (Phatthiyaphaibun et al., 2024); for Hindi, the Indic NLP Library (Kunchukuttan, 2020); and for Japanese, a tokenizer built on MeCab via the Fugashi wrapper (McCann, 2020; Kudo, 2006).

| Models | Train $\theta$ | Total $\theta$ | Dataset | en | es | hi | zh | $L_5$ | $L_{36}$ |
|--------|---------|---------|---------|------|------|------|------|------|------|
| mBLIP | 111M | 4.84B | 2.71M | **80.3** | 62.5 | 17.1 | 6.5 | 9.9 | 25.9 |
| PAELLA | 34M | 3B | 566K | 57.3 | 44.9 | 20.8 | 25.9 | 20.7 | 26.9 |
| BB + CC | 0.8B | 0.8B | 135M | 58.4 | 42.5 | 19.7 | 20.2 | **22.4** | 29.3 |
| Pangea | 7B | 7B | 6M | 75.9 | **64.6** | 16.2 | **29.0** | 12.5 | 31.8 |
| CONCAP | 111M | 4.84B | 566K | 72.4 | 58.6 | **24.4** | 21.7 | 18.2 | **34.2** |

Table 1: CIDEr scores on the XM3600 evaluation set, including results for four core languages (English, Spanish, Hindi, and Chinese), the average across five low-resource languages ($L_5$: Bengali, Māori, Quechua, Swahili, and Telugu), and the overall average across all 36 languages ($L_{36}$). We also report model size (Total $\theta$), the number of trainable parameters (Train $\theta$), and the size of the datasets used for training.

**Training Details** All models were trained with a learning rate of $3e^{-5}$ and a weight decay of 0.1. We applied Low-Rank Adaptation (LoRA) with rank 8, scaling factor $\alpha = 16$, and dropout rate 0.05, following the configuration used in mBLIP (Geigle et al., 2024) (see Table 3 in the Appendix for more training details.) Training was conducted for 10 epochs with epoch-based checkpointing using 2×A100 40GB GPUs and an effective batch size of 256. We selected the best checkpoint based on CIDEr scores on the COCO-35L development split (see Table 4 in Appendix for results). For all models, the final epoch gave the best results.

**Evaluation** We evaluate model performance using the widely established CIDEr metric (Vedantam et al., 2015), which measures the similarity between generated and reference captions based on n-gram overlap, placing greater weight on n-grams that are frequently agreed upon across references. For inference, we use beam search with a beam size of 5, a length penalty of 1, and a maximum output length of 25 tokens.

**Baselines** We compare our proposed method against several state-of-the-art multilingual image captioning models. Since CONCAP builds on top of mBLIP (Geigle et al., 2024), we begin by comparing it with the original mBLIP. Next, we compare it with PAELLA (Ramos et al., 2024), a multilingual retrieval-augmented model similar to ours, but with only retrieved captions. PAELLA uses CLIP-ViT-B/32 as encoder (Radford et al., 2021) and the multilingual XGLM (Lin et al., 2021) as decoder. We then compare with BB+CC (Thapliyal et al., 2022), a fully-supervised multilingual captioning model that combines the mT5-base (Xue et al., 2020) and ViT-B/16 models (Zhai et al., 2022), trained end-to-end on CC3M-35L and COCO-35L datasets. Finally, we compare against the recent Pangea (Yue et al., 2025), a large multilingual model with 7B trainable parameters based on LLaVA-Next architecture and trained on PangeaIns, a multilingual and culturally relevant multimodal dataset with 6 million instruction examples.

## 4.2 Main Results

Table 1 presents the main results, comparing CONCAP to several strong baselines on the XM3600 benchmark. We report CIDEr scores on the four core languages (English, Spanish, Hindi, and Chinese), the low-resource set $L_5$ (Bengali, Māori, Quechua, Swahili, and Telugu), and the full set of 36 languages ($L_{36}$), all as defined in (Thapliyal et al., 2022). We present results using language codes, with their mappings listed in Table 10.

**Averaged Performance ($L_{36}$)** We observe that, on average, CONCAP outperforms all baselines by a large margin, scoring 2.4 CIDEr points more than the next best model, Pangea. This is impressive given the large gap in trainable parameters (111M vs. 7B) and the amount of vision-language training data (566K vs. 6M data points). Comparing CONCAP to mBLIP, where both models share the same architecture and number of trainable parameters, we observe a gap of 8.3 CIDEr points, despite an almost five-fold reduction in training data with CONCAP. The CONCAP approach enables highly data-efficient training.

| Models | en | es | hi | zh | $L_5$ | $L_{36}$ |
|---|---|---|---|---|---|---|
| CONCAP | 72.4 | 58.6 | 24.4 | 21.7 | 18.2 | 34.2 |
| NoRAG | 66.0 | 48.9 | 20.4 | 17.4 | 17.2 | 26.9 |
| ConRAG | 70.9 | 55.0 | 19.5 | 20.9 | 17.1 | 30.3 |
| CapRAG | 66.2 | 53.3 | 23.9 | 20.2 | 16.9 | 31.4 |
| ConRAG$_{Rich}$ | 71.4 | 51.2 | 19.2 | 17.9 | 16.5 | 28.6 |
| CapRAG$_M$ | 38.3 | 30.4 | 16.4 | 13.9 | 13.1 | 20.4 |

Table 2: CIDEr scores on XM3600 across CONCAP and various ablations: NoRAG (no retrieval), CapRAG (caption retrieval), ConRAG (concept retrieval only), and two augmentations: CapRAG$_M$ (target language retrieval), and ConRAG$_{Rich}$ (extended concepts list).

**Per-language Performance**   Looking at individual languages, we see a more nuanced picture, where nearly every language ranks models differently. mBLIP appears to prioritize English, but is also strong on Spanish. Pangea ranks best on Spanish and Chinese but struggles on the five low-resource languages, where BB+CC, on the other hand, excels. CONCAP is best on Hindi and not particularly weak on any of the languages considered here. We show in §4.6 that CONCAP provides better coverage of low- and medium-resource languages, whereas Pangea offers better coverage of high-resource languages.

### 4.3   The Contribution of Concepts and Captions

Having established the strong overall performance of CONCAP, we now turn to an analysis of the individual contribution of its components, looking at models trained without retrieval augmentation (NoRAG), with retrieved captions (CapRAG), and retrieved concepts (ConRAG). Results are shown in Table 2.

The comparison to NoRAG isolates the strength of our proposed approach in an otherwise identical experimental setup. We find that CONCAP 's impressive performance is indeed attributed to the retrieval augmentation, as its non-augmented counterpart performs on par with weaker baselines from Table 1.

Looking at the two forms of retrieval augmentation in isolation, we find that each improves performance over NoRAG on average, with CapRAG being slightly more effective than ConRAG. While the gain from caption retrieval corroborates prior work (Ramos et al., 2024), it is interesting to see that concepts alone can also provide a highly effective signal.

The most insightful finding here is that CONCAP considerably outperforms both ConRAG and CapRAG, indicating that the gains from these two forms of retrieval augmentation are *additive*: captions help guide generation with fluent language patterns, while concepts ensure broader content coverage and more accurate visual grounding.

### 4.4   The Role of English as a Pivot

Previous work has demonstrated that retrieving captions in English yields good results in multilingual settings, as shown in COCO-35L. However, this exploits a particularity of the COCO-35L dataset, while in practice, English translations might not be available. This motivates exploring retrieval strategies that work directly in each target language.

We thus experiment with retrieval directly in the target language (using mSigLIP) at training time.[1] We carry out this experiment in the CapRAG setting and report results in Table 2 (CapRAG$_M$). CapRAG$_M$ considerably underperforms CapRAG across all individual languages and language groupings and results in worse performance even than NoRAG, i.e., in this scenario, caption retrieval actively hurts performance. This indicates that the quality of the retriever is of utmost importance for the success of *caption-based* retrieval augmentation in multilingual image captioning. In this sense, *concept-based* retrieval augmentation

---

[1]Recall that at inference time, retrieval is always done in the target language (see §4.1.)

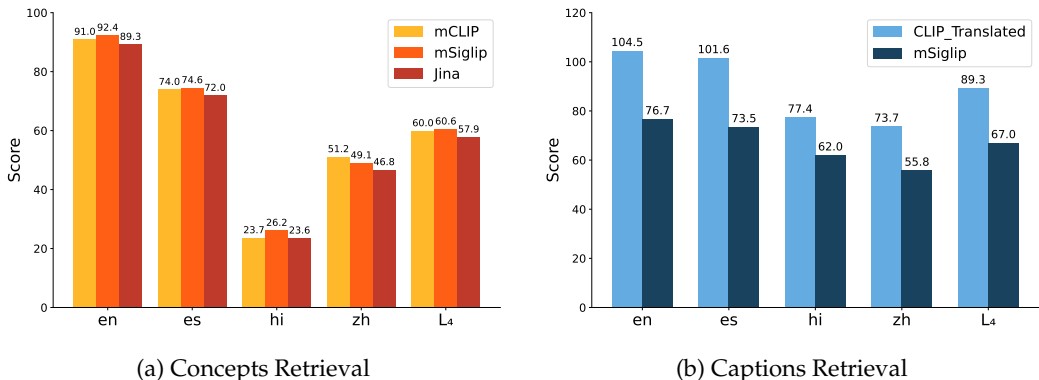

(a) Concepts Retrieval        (b) Captions Retrieval

Figure 2: Retrieval performance on the COCO development set across two settings: (a) compares mCLIP, mSigLIP, and Jina on retrieved concepts, (b) compares English CLIP retrieval against mSigLIP on retrieved captions.

(ConRAG) offers a more reliable alternative, as it does not depend on translation and aids performance regardless of possible shortcomings, i.e., noise, in the retrieval process.

### 4.5 Best Retriever

To identify the most effective multilingual vision-language representation model for concept retrieval, we conducted a comparative study of three widely-used models: mSigLip, mCLIP, and Jina-CLIP (Zhai et al., 2023; Chen et al., 2023; Koukounas et al., 2024). For faster iteration, we used a reduced training set of 113k samples, equally distributed across languages. We trained a ConRAG model for 5 epochs, using retrieved concepts from each of the three retrievers, and evaluated performance on the COCO development split. As shown in Figure 2a, mSigLIP achieved the highest performance on average, and was therefore selected as the default multilingual retriever for all our experiments.

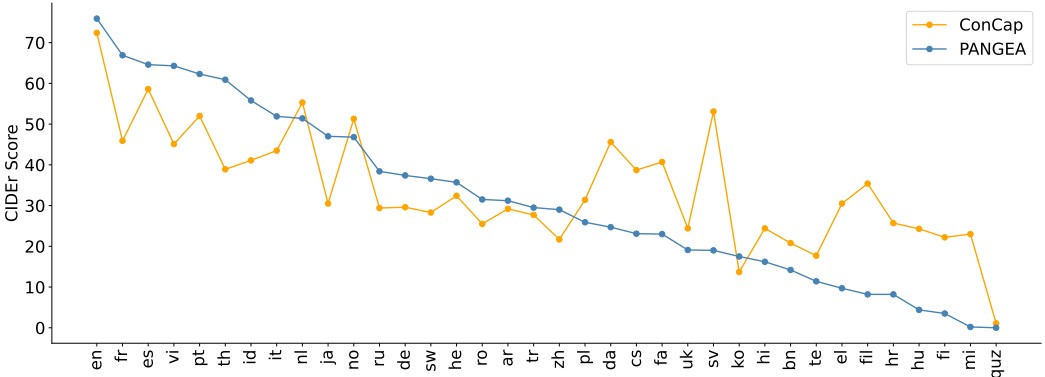

Figure 3: Per-language performance between CONCAP and Pangea on XM3600.

### 4.6 Per-language Performance

In Figure 3 we provide a breakdown of performance per language, comparing CONCAP and the next-best model from Table 1, Pangea. The languages are arranged in descending order based on their Pangea scores. We observe that CONCAP consistently outperforms Pangea across many low- and mid-resource languages. Notable gains are observed in languages such as Telugu, Hindi, Māori, Filipino, Farsi, and others. Meanwhile, Pangea performs strongly on high-resource languages such as English, Spanish, and Portuguese, reinforcing the divide between high- and low-resource languages even further.

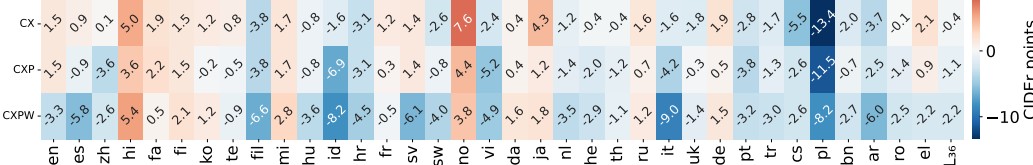

Figure 4: Performance difference (Δ CIDEr) on the XM100 test set when changing the wordlists: COCO-35L+XM3600* (CX), CX+Pangea (CXP), CXP+Wikipedia (CXPW). The * indicates that XM3600 has been filtered to exclude any words and images related to the test subset (XM100) to avoid contamination. Positive values indicate improvement over the COCO baseline; negative values indicate a drop.

## 5 Concept Enrichment

The concept retrieval, in its base form discussed above, is done using a concept list built from the training data, which, as a reminder, is translated from English and grounded in Western images. The natural next step is the integration of culturally-representative lexicons. Enriched concept retrieval could result in better generalization and out-of-domain coverage. Below, we present early results in this direction showing that this is not a trivial task.

We enrich the default COCO-35L concept lists for each language with additional concepts sourced from the PangeaIns dataset (Yue et al., 2025). Specifically, we focus on the *cultural* portion of the data, aiming to accommodate the geographic diversity of XM3600 with relevant cultural concepts. The results in Table 2 (ConRAG$_{Rich}$) indicate a negative impact from concept enrichment: performance drops by 1.7 CIDEr points relative to ConRAG. This suggests that adding more terms from external datasets may introduce noisy, less relevant concepts, which dilute the retrieval quality.

To better understand this finding, we test ConRAG on XM100 with varying concept list configurations, focusing on sensitivity to datastore size and makeup. We compare: (1) **CX**, which adds filtered XM3600 lexicons (excluding the XM100 captions); (2) **CXP**, which includes PangeaIns cultural terms; and (3) **CXPW**, which adds Wikipedia and Common Crawl entries for broader but less focused coverage. Per-language results are presented in Figure 4, as a change in CIDEr score from the base lexicon (COCO-35L) to the different augmented lexicon configurations.

This addition of the filtered XM3600 lexicon (CX) shows a mixed effect, with roughly half of the languages seeing improvements and the other half experiencing a performance drop. Expanding this setup with a culturally-relevant lexicon (CXP) leads to a further decline in performance for a larger portion of the languages. Finally, incorporating a broad web-based lexicon (CXPW) results in the majority of languages showing a degradation in performance. While expanding the lexicon pool could, in theory, improve the coverage of retrieved concepts with geographically diverse and culturally relevant terms, in practice, it seems to add noise, which distracts the generation process and yields lower-quality captions.

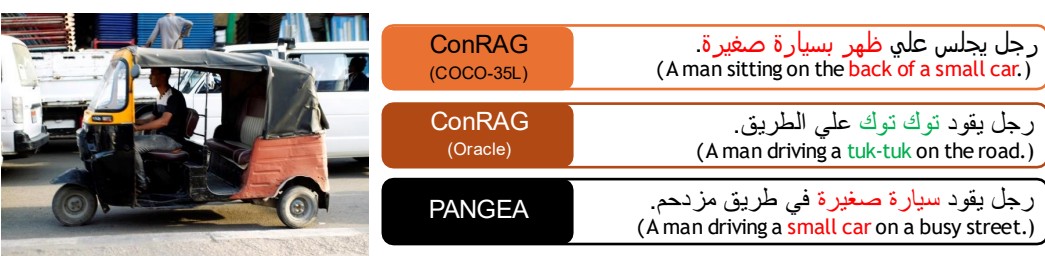

Figure 5: Oracle-augmented ConRAG correctly identifies the tuk-tuk in the image.

Lastly, we conduct an oracle experiment on 200 images from the JEEM dataset for cross-cultural Arabic image captioning (Kadaoui et al., 2025). The subset consists of highly cultural images, manually selected by a native Arabic speaker based solely on visual content, and manually annotated with highly relevant cultural concepts. The ConRAG model, augmented with these oracle concepts, achieves a CIDEr score of **17.9**, compared to **13.9** when retrieving concepts from a general-purpose concept list (COCO-35L). A qualitative example of improved cultural awareness is shown in Figure 5. These gains can be observed only in the early stages of training (the results above correspond to one epoch of training). Continued finetuning seems to shift the decoder's output distribution, making unseen lexical items harder to predict, even when prompted.

## 6 Conclusion

We introduced CONCAP, a multilingual image captioning model that enhances caption generation by integrating retrieved captions with image-specific concepts. This approach improves caption quality while reducing the need for extensive multilingual training. Experiments on XM3600 show that CONCAP outperforms mBLIP and Pangea despite using fewer training resources. Ablation studies confirm the additive benefits of caption and concept retrieval. Further analysis informs the choice of retriever and highlights the brittleness of standard caption retrieval strategies. Concepts retrieved from enriched lexicons prove to be noisier and to hinder the accurate generation of image captions. Oracle experiments with cultural concepts show that this form of retrieval augmentation improves captioning initially, but the benefits fade as training progresses: the model increasingly depends on its learned vocabulary, limiting integration of new cultural concepts. Future work should investigate more effective ways for concept enrichment and consider targeting datasets with more pronounced cultural representation, where well-informed concept enrichment may prove more effective.

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

# A Appendix

## A.1 Training Configuration and Checkpoint Selection

The training configuration detailed in Table 3 indicates that we consistently used a batch size of 256 for all experiments. Our models were trained using 2 A100s 40G GPUs, with the CONCAP taking approximately 66 hours to complete 10 epochs.

| Model | Batch Size | Grad Accum | Seq Len | Eff. Batch | Time (hrs) |
|---|---|---|---|---|---|
| Normal | 64 | 2 | 8 | 256 | 38 |
| ConRAG | 32 | 4 | 56 | 256 | 45 |
| CapRAG | 16 | 8 | 111 | 256 | 58 |
| CONCAP | 16 | 8 | 155 | 256 | 66 |

Table 3: Training configurations for different models. All experiments used $2\times$ A100 40GB GPUs and ran for 10 epochs with checkpointing.

Table 4 presents the CIDEr scores for each checkpoint on the COCO-35L development set. It is evident that the final checkpoint achieved the highest score for all models.

| Model | 1 | 2 | 3 | 4 | 5 | 6 | 7 | 8 | 9 | 10 |
|---|---|---|---|---|---|---|---|---|---|---|
| NoRAG | 66.3 | 78.6 | 85.1 | 89.8 | 92.9 | 95.9 | 97.3 | 98.2 | 98.8 | 99.2 |
| CapRAG | 78.6 | 89.6 | 92.8 | 94.1 | 95.3 | 97.0 | 99.1 | 100.3 | 101.2 | 102.3 |
| ConRAG | 66.3 | 76.2 | 81.4 | 83.7 | 85.1 | 88.7 | 92.0 | 92.8 | 93.4 | 95.1 |
| CONCAP | 79.3 | 92.7 | 97.5 | 99.9 | 101.2 | 102.7 | 105.5 | 107.3 | 108.2 | 109.4 |

Table 4: Average CIDEr scores on the COCO-35L dev split across four core languages for each checkpoint.

## A.2 Prompts For Concept Retrieval

In Table 9, we present the multilingual prompts used for concept retrieval. Initially, we translated the prefix from English to other languages using GPT-4o (OpenAI, 2024). These translations were later verified with Google Translate to ensure semantic consistency and linguistic accuracy. For Chinese and Japanese, an additional suffix was required to add nuance to the sentence and preserve natural phrasing.

## A.3 Number of Concepts

We conducted an ablation study on the COCO development set to determine the ideal number of retrieved concepts. ConRAG was trained on 20% of the data, and we evaluated different numbers of shots on the COCO validation split. As shown in Table 5, performance for 10 and 20 retrieved concepts is virtually identical, but augmenting generation with more retrieved concepts increases the inference cost. Accordingly, we chose to use $m = 10$.

| $m$ | en | es | hi | zh | $L_4$ |
|---|---|---|---|---|---|
| 4 | 90.4 | 69.1 | 26.6 | 47.9 | 58.5 |
| 10 | 92.4 | 74.6 | 26.2 | 49.2 | 60.6 |
| 20 | 92.5 | 77.1 | 24.2 | 47.0 | 60.2 |

Table 5: Captioning performance by language across 4, 10, and 20 retrieved concepts.

### A.4 Memory Usage

Table 9 reports the COCO concept list length for each language, with an average of approximately 43,940 words. Given that MS-COCO contains 566,000 captions, the memory footprint of the concepts datastore is less than 10 percent of that of the captions datastore. A visual comparison of the caption and concept datastore sizes across languages is shown in Figure 6. Since concepts and captions can be retrieved in parallel, with the newly added concept datastore being smaller and thus faster to access, this addition introduces no latency to the retrieval process.

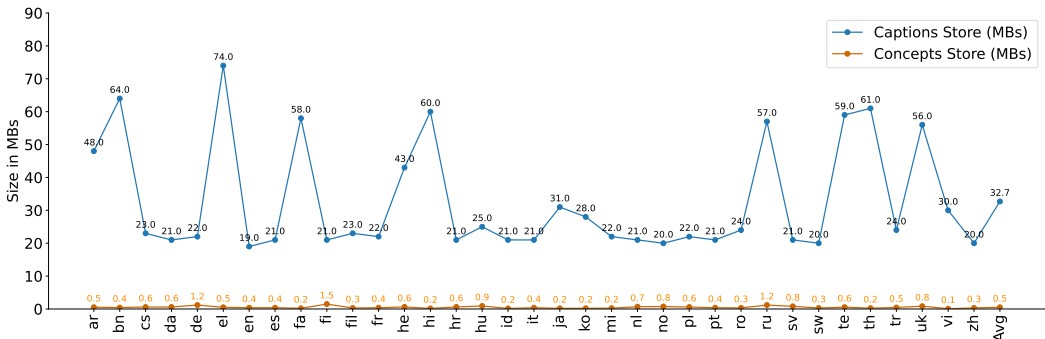

Figure 6: Comparison between the file sizes of the caption and the concepts datastores per language.

### A.5 Per-Language Performance

Figure 7 details per-language performance for CONCAP, ConRAG, and CapRAG. We see that CONCAP consistently outperforms the other two variants on all languages except for Maori (mi), which seems to benefit more from retrieved concepts than captions.

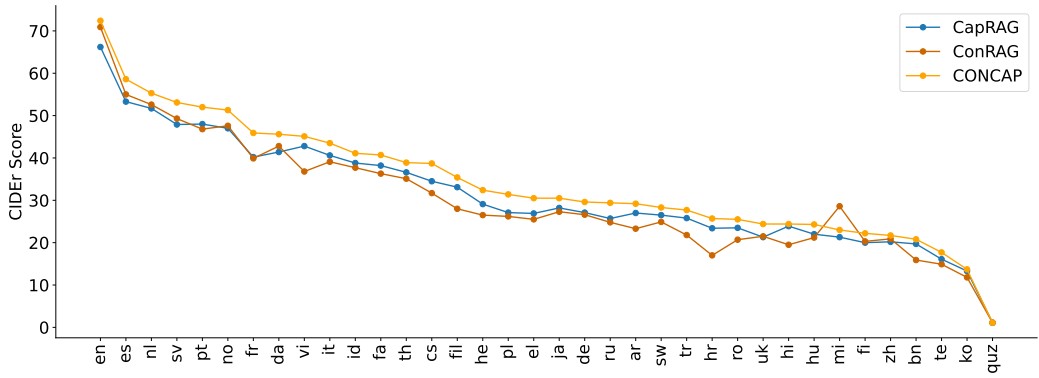

Figure 7: Per-Language performance between CONCAP, ConRAG, and CapRAG.

### A.6 Concept List Sizes and Performance Scores

Table 9 shows the sizes of the concept lists for each language across different dataset combinations. We began with the COCO-35L words, then filtered the XM3600 to exclude the image-caption pairs present in XM100 to prevent leakage. After that, we included the remaining words. Next, we incorporated the PangeaIns cultural subset to add more words not found in the COCO-35L translated data. Finally, we appended words from the Fasttext library, which was trained on Wikipedia and Common Crawl.

### A.7 Results on All Languages

Table 10 presents the CIDEr scores across all languages, while Table 11 shows the BLEU scores (Papineni et al., 2002), both evaluated on the XM3600 dataset. We compare the performance of all our approaches against the state-of-the-art model, Pangea (Yue et al., 2025). $L_4$ refers to the core languages English (en), Spanish (es), Chinese (zh), and Hindi (hi), as defined in prior work (Thapliyal et al., 2022).

### A.8 Linguistic Prior

Retrieval-augmented generation (RAG) introduces a linguistic prior that, in our context, functions as a complementary signal rather than a substitute for visual grounding. To assess this, we include two key baselines in our experiments: NoRAG, which leverages only visual inputs, and CONCAP, which integrates both image and retrieved textual information. Additionally, we conduct a Text-only ablation in which the original image is replaced with a solid-color version computed from its mean RGB value, effectively removing all visual content while preserving layout and input structure. This ablation is designed to isolate the model's dependence on the retrieved text. Preliminary results, summarized in Table 6, indicate that CONCAP consistently outperforms both NoRAG and Text-only across the four primary languages. These findings suggest that the model benefits from a combination of modalities and does not exhibit an overreliance on the retrieved text alone.

| Model | English | Spanish | Chinese | Hindi | $L_4$ |
|---|---|---|---|---|---|
| NoRAG | 66.0 | 48.9 | 17.4 | 20.4 | 38.2 |
| Text-only | 29.1 | 9.6 | 4.11 | 10.8 | 13.4 |
| CONCAP | 72.4 | 58.6 | 21.7 | 24.4 | **44.3** |

Table 6: Comparison between Image-only (NoRAG), Text-only, and CONCAP across the four core languages

| Lang | COCO | +XM3600* | +Pangea | +Wiki |
|------|------|----------|---------|-------|
| en | 27456 | 27878 | 27878 | 204316 |
| es | 28187 | 29193 | 101099 | 223659 |
| zh | 34971 | 37622 | 153658 | 270470 |
| hi | 13502 | 13891 | 17546 | 24723 |
| ar | 52423 | 55558 | 138879 | 235593 |
| fr | 28630 | 30179 | 100639 | 226474 |
| de | 76307 | 81124 | 257457 | 354984 |
| ru | 67761 | 70790 | 249805 | 331528 |
| ja | 18375 | 20109 | 71197 | 71197 |
| pt | 27150 | 28635 | 104681 | 225965 |
| it | 27150 | 31624 | 118497 | 234006 |
| nl | 55286 | 57027 | 83447 | 241859 |
| tr | 49688 | 51533 | 201970 | 294533 |
| sv | 68342 | 70449 | 70449 | 244036 |
| no | 68342 | 68561 | 275367 | 391581 |
| da | 64403 | 66429 | 66429 | 239378 |
| fi | 116243 | 120918 | 120918 | 282930 |
| pl | 65313 | 67677 | 121040 | 243183 |
| cs | 64368 | 65982 | 104444 | 234808 |
| uk | 65293 | 69289 | 116724 | 245061 |
| ro | 65293 | 39233 | 140965 | 254239 |
| el | 40270 | 43154 | 83401 | 227733 |
| he | 50846 | 53321 | 101299 | 212993 |
| bn | 23825 | 24431 | 27381 | 32391 |
| fa | 17555 | 19612 | 29802 | 170328 |
| te | 38794 | 39586 | 40964 | 193375 |
| sw | 34741 | 37141 | 122410 | 276114 |
| vi | 8597 | 8943 | 39866 | 214706 |
| th | 6414 | 8114 | 22497 | 40005 |
| id | 18734 | 19921 | 88607 | 229330 |
| ko | 21221 | 30353 | 346315 | 346315 |
| hu | 90319 | 94465 | 94465 | 257909 |
| hr | 56070 | 58227 | 58227 | 221033 |
| fil | 32680 | 34503 | 34503 | 211716 |
| mi | 13352 | 14799 | 14799 | 14799 |

Table 9: Size of the concept list for each language across different dataset combinations.

| Language | Prompt Format |
|----------|---------------|
| ar | صورة لـ <concept> |
| bn | একটি ছবির <concept> |
| cs | fotografie z <concept> |
| da | et foto af en <concept> |
| de | ein Foto von ein <concept> |
| el | μια φωτογραφία από <concept> |
| en | a photo of a <concept> |
| es | una foto de una <concept> |
| fa | عکس از یک <concept> |
| fi | valokuva <concept> |
| fil | isang larawan ng <concept> |
| fr | une photo de <concept> |
| he | תמונה של <concept> |
| hi | एक तस्वीर की <concept> |
| hr | fotografija od <concept> |
| hu | egy fotó a <concept> |
| id | sebuah foto dari <concept> |
| it | una foto di <concept> |
| ja | 一枚の<concept>の写真 |
| ko | 한 장의 사진 <concept> |
| mi | he whakaahua o te <concept> |
| nl | een foto van een <concept> |
| no | et bilde av en <concept> |
| pl | zdjęcie <concept> |
| pt | uma foto de <concept> |
| quz | huk fotota a <concept> |
| ro | o fotografie a unui <concept> |
| ru | фотография <concept> |
| sv | ett foto på en <concept> |
| sw | picha ya <concept> |
| te | ఒక ఫోటో ఒక <concept> |
| th | รูปถ่ายของ <concept> |
| tr | bir fotoğrafı <concept> |
| uk | фотографія a <concept> |
| vi | một bức ảnh của một <concept> |
| zh | 一张<concept>的照片 |

Table 9: Language-specific prompt templates used for concept retrieval. Each row shows the prefix and, where applicable, the suffix in Chinese and Japanese used to wrap a concept token in a grammatically correct and semantically natural form for that language.

| Models | en | es | zh | hi | fa | fi | ko | te | fil | mi |
|---|---|---|---|---|---|---|---|---|---|---|
| mBLIP | 80.3 | 62.5 | 17.1 | 6.5 | 4.2 | 16.9 | 11.1 | 11.9 | 19.2 | 9.5 |
| Pangea | 75.9 | 64.6 | 16.2 | 29.0 | 23.0 | 3.5 | 17.5 | 11.4 | 8.2 | 0.2 |
| NoRAG | 66.0 | 48.9 | 20.4 | 17.4 | 28.9 | 16.6 | 8.7 | 15.0 | 25.1 | 30.8 |
| ConRAG | 70.9 | 55.0 | 19.5 | 20.9 | 36.3 | 20.3 | 11.8 | 14.9 | 28.0 | 28.6 |
| ConRAG$_{Rich}$ | 71.4 | 51.2 | 19.2 | 17.9 | 32.1 | 19.1 | 11.2 | 14.4 | 25.7 | 29.5 |
| CapRAG | 66.0 | 50.6 | 23.2 | 19.8 | 35.8 | 19.2 | 11.9 | 22.8 | 37.3 | 44.8 |
| CapRAG$_M$ | 66.2 | 53.3 | 23.9 | 20.2 | 38.2 | 20.0 | 13.3 | 16.1 | 33.1 | 21.3 |
| CONCAP | 72.4 | 58.6 | 24.4 | 21.7 | 40.7 | 22.2 | 13.7 | 17.7 | 35.4 | 23.0 |

| Models | hu | id | hr | fr | sv | sw | no | vi | da |
|---|---|---|---|---|---|---|---|---|---|
| mBLIP | 21.5 | 38.7 | 4.9 | 58.4 | 48.7 | 14.3 | 46.3 | 35.8 | 4.3 |
| Pangea | 4.4 | 55.8 | 8.2 | 66.9 | 19.0 | 36.6 | 46.8 | 64.3 | 24.7 |
| NoRAG | 18.6 | 33.8 | 13.1 | 36.7 | 42.6 | 22.3 | 43.1 | 31.8 | 37.7 |
| ConRAG | 21.2 | 37.7 | 17.0 | 39.9 | 49.3 | 24.9 | 47.6 | 36.8 | 42.8 |
| ConRAG$_{Rich}$ | 21.7 | 34.5 | 15.0 | 37.6 | 47.7 | 21.6 | 45.9 | 32.2 | 41.5 |
| CapRAG | 20.8 | 36.9 | 22.4 | 38.1 | 44.4 | 26.5 | 47.0 | 42.8 | 41.4 |
| CapRAG$_M$ | 22.0 | 38.8 | 23.4 | 40.2 | 47.9 | 18.7 | 27.2 | 32.9 | 25.6 |
| CONCAP | 24.3 | 41.1 | 25.7 | 45.9 | 53.1 | 28.3 | 51.3 | 45.1 | 45.6 |

| Models | ja | nl | he | th | ru | it | uk | de | pt | tr |
|---|---|---|---|---|---|---|---|---|---|---|
| mBLIP | 0.2 | 55.3 | 18.5 | 0.3 | 26.6 | 45.8 | 20.3 | 31.6 | 54.1 | 24.0 |
| Pangea | 47.0 | 51.4 | 35.7 | 60.9 | 38.4 | 51.9 | 19.1 | 37.4 | 62.3 | 29.5 |
| NoRAG | 23.6 | 47.4 | 18.9 | 34.0 | 22.3 | 31.5 | 18.5 | 24.8 | 45.8 | 18.3 |
| ConRAG | 27.3 | 52.6 | 26.5 | 35.1 | 24.8 | 39.1 | 21.5 | 26.6 | 46.8 | 21.8 |
| ConRAG$_{Rich}$ | 23.2 | 50.7 | 23.8 | 33.0 | 23.3 | 34.5 | 19.9 | 27.8 | 45.1 | 21.4 |
| CapRAG | 28.2 | 51.7 | 29.1 | 36.6 | 25.7 | 40.6 | 21.3 | 27.1 | 48.0 | 25.8 |
| CapRAG$_M$ | 21.3 | 28.6 | 18.8 | 25.3 | 16.9 | 27.3 | 14.9 | 17.6 | 29.7 | 17.1 |
| CONCAP | 30.5 | 55.3 | 32.4 | 38.9 | 29.4 | 43.5 | 24.4 | 29.6 | 52.0 | 27.7 |

| Models | cs | pl | bn | ar | ro | el | quz | L$_4$ | L$_5$ | L$_{36}$ |
|---|---|---|---|---|---|---|---|---|---|---|
| mBLIP | 29.6 | 30.9 | 12.9 | 21.3 | 22.6 | 23.8 | 1.1 | 41.6 | 9.9 | 25.9 |
| Pangea | 23.1 | 25.9 | 14.2 | 31.2 | 31.5 | 9.7 | 0.0 | 46.4 | 12.5 | 31.8 |
| NoRAG | 28.0 | 22.0 | 16.6 | 17.9 | 17.3 | 21.8 | 1.1 | 38.2 | 17.2 | 26.9 |
| ConRAG | 31.7 | 26.2 | 15.9 | 23.3 | 20.7 | 25.5 | 1.1 | 41.6 | 17.1 | 30.3 |
| ConRAG$_{Rich}$ | 29.2 | 25.4 | 16.0 | 22.0 | 19.5 | 23.0 | 1.1 | 39.9 | 16.5 | 28.6 |
| CapRAG | 34.5 | 27.1 | 19.7 | 27.0 | 23.5 | 26.9 | 1.1 | 40.9 | 16.9 | 31.4 |
| CapRAG$_M$ | 19.9 | 17.3 | 15.4 | 17.0 | 16.3 | 16.8 | 1.1 | 24.8 | 13.1 | 20.4 |
| CONCAP | 38.7 | 31.4 | 20.8 | 29.2 | 25.5 | 30.5 | 1.1 | 44.3 | 18.2 | 34.2 |

*Language codes:* en=English, es=Spanish, zh=Chinese, hi=Hindi, fa=Farsi, fi=Finnish, ko=Korean, te=Telugu, fil=Filipino, mi=Maori, hu=Hungarian, id=Indonesian, hr=Croatian, fr=French, sv=Swedish, sw=Swahili, no=Norwegian, vi=Vietnamese, da=Danish, ja=Japanese, nl=Dutch, he=Hebrew, th=Thai, ru=Russian, it=Italian, uk=Ukrainian, de=German, pt=Portuguese, tr=Turkish, cs=Czech, pl=Polish, bn=Bengali, ar=Arabic, ro=Romanian, el=Greek, quz=Quechua.

Table 10: CIDEr Scores: XM3600 Evaluation on 36 languages.

| Models | en | es | zh | hi | fa | fi | ko | te | fil | mi |
|---|---|---|---|---|---|---|---|---|---|---|
| NoRAG | 11.7 | 8.0 | 2.1 | 2.8 | 4.7 | 1.2 | 0.4 | 0.7 | 7.1 | 8.5 |
| ConRAG | 11.7 | 8.5 | 2.1 | 2.5 | 5.0 | 1.2 | 0.4 | 0.7 | 6.8 | 8.2 |
| ConRAG$_{Rich}$ | 11.9 | 8.0 | 1.7 | 2.4 | 4.7 | 1.2 | 0.4 | 0.6 | 6.6 | 8.4 |
| CapRAG | 11.0 | 8.5 | 1.9 | 2.6 | 5.2 | 1.3 | 0.6 | 0.5 | 7.8 | 7.6 |
| CapRAG$_M$ | 5.5 | 4.4 | 1.0 | 1.6 | 3.3 | 0.6 | 0.3 | 0.0 | 4.7 | 5.8 |
| CONCAP | 11.9 | 9.5 | 2.0 | 2.9 | 5.9 | 1.4 | 0 | 0.5 | 8.4 | 8.1 |

| Models | hu | id | hr | fr | sv | sw | no | vi | da |
|---|---|---|---|---|---|---|---|---|---|
| NoRAG | 1.9 | 5.1 | 1.2 | 7.1 | 6.7 | 2.6 | 8.1 | 9.5 | 6.7 |
| ConRAG | 2.0 | 5.1 | 1.2 | 7.1 | 7.2 | 2.7 | 8.3 | 9.7 | 6.7 |
| ConRAG$_{Rich}$ | 2.2 | 4.6 | 1.2 | 6.6 | 6.8 | 2.5 | 8.0 | 9.3 | 6.4 |
| CapRAG | 2.0 | 5.0 | 2.4 | 7.4 | 7.1 | 3.0 | 7.6 | 10.4 | 6.3 |
| CapRAG$_M$ | 1.0 | 3.2 | 1.4 | 4.9 | 3.4 | 1.9 | 4.0 | 7.9 | 3.7 |
| CONCAP | 2.5 | 5.5 | 2.8 | 8.6 | 7.9 | 3.2 | 8.7 | 11.0 | 7.3 |

| Models | ja | nl | he | th | ru | it | uk | de | pt | tr |
|---|---|---|---|---|---|---|---|---|---|---|
| NoRAG | 7.2 | 8.8 | 2.1 | 6.6 | 2.5 | 5.6 | 2.3 | 3.9 | 7.9 | 2.1 |
| ConRAG | 7.8 | 8.9 | 2.4 | 7.1 | 2.5 | 6.3 | 2.4 | 3.4 | 7.6 | 1.9 |
| ConRAG$_{Rich}$ | 6.7 | 8.7 | 2.1 | 6.9 | 2.2 | 5.7 | 2.2 | 3.8 | 7.2 | 1.9 |
| CapRAG | 7.8 | 9.0 | 3.0 | 6.3 | 2.7 | 6.7 | 2.2 | 3.8 | 8.2 | 2.3 |
| CapRAG$_M$ | 5.2 | 4.1 | 1.7 | 3.0 | 1.6 | 4.1 | 1.4 | 2.4 | 4.4 | 1.4 |
| CONCAP | 10.6 | 9.3 | 3.3 | 6.8 | 3.4 | 7.3 | 2.9 | 4.3 | 8.9 | 2.5 |

| Models | cs | pl | bn | ar | ro | el | quz | L$_4$ | L$_5$ | L$_{36}$ |
|---|---|---|---|---|---|---|---|---|---|---|
| NoRAG | 3.1 | 2.7 | 0.6 | 1.6 | 4.3 | 1.8 | 0.0 | 6.2 | 2.5 | 4.4 |
| ConRAG | 2.8 | 3.0 | 0.5 | 2.0 | 4.5 | 1.7 | 0.0 | 6.2 | 4.5 | 2.4 |
| ConRAG$_{Rich}$ | 2.5 | 2.9 | 0.5 | 1.8 | 4.3 | 1.5 | 0.0 | 6.0 | 4.3 | 2.4 |
| CapRAG | 3.2 | 3.2 | 0.5 | 2.5 | 4.8 | 1.8 | 0.0 | 6.0 | 2.3 | 4.6 |
| CapRAG$_M$ | 1.5 | 1.7 | 0.3 | 1.3 | 2.7 | 1.0 | 0.0 | 3.1 | 1.6 | 2.7 |
| CONCAP | 4.0 | 3.8 | 0.5 | 2.6 | 5.2 | 2.1 | 0.0 | 6.6 | 2.5 | 5.1 |

*Language codes:* en=English, es=Spanish, zh=Chinese, hi=Hindi, fa=Farsi, fi=Finnish, ko=Korean, te=Telugu, fil=Filipino, mi=Maori, hu=Hungarian, id=Indonesian, hr=Croatian, fr=French, sv=Swedish, sw=Swahili, no=Norwegian, vi=Vietnamese, da=Danish, ja=Japanese, nl=Dutch, he=Hebrew, th=Thai, ru=Russian, it=Italian, uk=Ukrainian, de=German, pt=Portuguese, tr=Turkish, cs=Czech, pl=Polish, bn=Bengali, ar=Arabic, ro=Romanian, el=Greek, quz=Quechua.

Table 11: BLEU scores: XM3600 Evaluation on 36 languages.

