# OpenReview forum: "CONCAP: Seeing Beyond English with Concepts Retrieval-Augmented Captioning"
_colmweb.org/COLM/2025/Conference — COLM 2025_

### Official Review · Reviewer_H9Ar · 2025-05-12

**Rating:** 7
**Confidence:** 2
**Ethics Flag:** 1

**Summary:**

The paper presents CONCAP, a parameter-efficient* architecture for generative multilingual image captioning based on retrieval-augmented generation, with retrieval of captions over a datastore of images and of concepts over a datastore of terms. The authors instantiate their architecture over an mBLIP image captioning model, training it over the dataset COCO-35, and evaluating it on XM3600 and the subset XM100. The presented model achieves good performance overall, with remarkable improvement on low-resource languages compared to the base mBLIP model, while maintaining good performance in other languages, surpassing the overall performance of other multilingual models, such as PANGEA and PAELLA. The authors present substantial evaluations, along with clarifying ablation studies investigating different methods for retrieval and their impacts on the performance.


It is important to notice that the model is in some sense over-reliant on the English language, as shown in their results, in the sense that, by training on a dataset constituted of English captions translated into other languages, linguistic and cultural biases in descriptions may be learned by the model, which may be reproduced in the generated captions during evaluation. In fact, it is not completely clear if a multilingual and multicultural image-language dataset can be consistent, in the terms proposed by Thapliyal et al in the proposal of XM3600, since, as discussed for example by Liu et al. [1], people from different cultures may evoke different concepts to describe scenes (e.g. a bowl of fruits can be seen as 'a bowl of exotic fruits' or  'an offering' depending on cultural and religious background). As such, by training on COCO-35 and evaluating on XM3600 the authors may underrepresent the performance of their model. In fact, we can see some variation in the description of the images in different languages on XM3600. It would be an interesting exercise to train the model on part of XM3600 and test on XM100, for example.

There are some typos in the text and duplicate entries in the bibliography that should be revised (listed below):
 - several citations in the text should not be direct citations Author (year) =, but indirect citations (author, year), as the authors are not directly referencing the work, only talking about it (page 4, base model -Hu et al;  page 6, baselines - several)
 - section 4.2: should it be L_{36} instead of L_{35}?
 - what is L_4 in Tables 7 and 8? It has not been introduced
 - Ramos et al 2023 (LMCap) appears twice - keep the peer-reviewed version
 - Liu et al 2023 appears twice - keep the peer-reviewed version


* In the sense that only a part of the architecture is trained end-to-end, thus only a fraction of the overall parameters are trainable, as it takes advantage of pretrained image-encoders and language-decoders.

[1] LIU, Fangyu et al. Visually Grounded Reasoning across Languages and Cultures. In: Proceedings of the 2021 Conference on Empirical Methods in Natural Language Processing. 2021. p. 10467-10485.

**Questions To Authors:**

- Why the two different dataset chosen for training and evalutation, with training over a translation-based dataset? Did the authors consider using two manually annotated and culturally diverse datasets, such as XM3600 and PageaIns for training/evaluation? How are the results expected to vary in this scenario?

- Why in the investigation of the use of English as pivot, the authors compare the results of the model with CLIP (CapRAG) and with mSigLIP (CapRAG_M), instead of using msiglip in both cases? How can we compare both results and achieve any meaningful conclusion on the role of English as pivot in the experiment if the encoder has changed?

- Since the size of the concept list seems to have no direct correlation with the improvement in performance of the model, how can we evaluate the contribution of the concept lists? Why all unique tokens are considered valid concepts, when most tokens cannot be directly related to concepts? From the concept list, in the evaluation experiments, which concepts have been more often retrived and how do they relate to the images (this evaluation could be feasibly be done for the XM100 dataset)? this distribuition has changed considerably in the experiments with different concept lists? how? It would be interesting for future work to validate or order concept candidates by importance - e.g. through notions such as lexical salience [1] (often related to surprisal or predictability).


[1] Erkan, G., & Radev, D. R. (2004). Lexrank: Graph-based lexical centrality as salience in text summarization. Journal of artificial intelligence research, 22, 457-479.

**Reasons To Accept:**

- A parameter-efficient architecture that is easily adaptable for new and improved image encoders and language decoders, trained and evaluated on a multitude of typologically diverse languages with competing results;

- The work investigates the use of concept-based augmentation in image captioning, which to my knowledge, is new.

**Reasons To Reject:**

- While the work is interesting and novel in the sense of incorporating concept augmented retrieval to the problem, conceptually it can be seen as an incremental contribution. If we compare it to PAELLA - a much smaller model - it is not clear if the increased performance is due to the concept augmentation of simply as a function or the increased number of trainable parameters in the  model.

---

> ### Author Response · Authors · 2025-06-01
>
> We thank the reviewer for the detailed feedback. We acknowledge the typos, citation formatting issues, and bibliography duplications, and we will correct all of them in the camera-ready version.
>
> > ... it is not clear if the increased performance is due to the concept augmentation of simply as a function or the increased number of trainable parameters in the model.
>
> This concern can be resolved with reference to the ablations in Table 2 where we see that for a controlled number of trainable parameters, retrieval augmentated generation with concepts and captions (ConCap) outperforms caption RAG alone (CapRAG) and concept RAG alone (CapRAG).
>
> We address below the questions raised by the reviewer:
>
> > Why are two different dataset chosen for training and evalutation, with training over a translation-based dataset? Did the authors consider using two manually annotated and culturally diverse datasets, such as XM3600 and PageaIns for training/evaluation? How are the results expected to vary in this scenario?
>
> The pairing of COCO-35L for training and XM3600 for evaluation is standard in work on multilingual image captioning. XM3600 is designed exclusively as a test set, while PangeaIns is not a manually annotated dataset (it is a synthetic dataset generated using GPT4) and is a conversational dataset rather than an image captioning one.As such, neither XM3600 nor PangeaIns would have been suitable alternatives to COCO-35L training.
>
> >  Why in the investigation of the use of English as pivot, the authors compare the results of the model with CLIP (CapRAG) and with mSigLIP (CapRAG_M), instead of using msiglip in both cases? How can we compare both results and achieve any meaningful conclusion on the role of English as pivot in the experiment if the encoder has changed?
>
> In preliminary experiments with mSigLIP and CLIP used for retrieval in the English-as-a-pivot setting, we discovered that CLIP is considerably better suited for this setting (the results can be seen here: https://imgur.com/a/Rya5nGx). For this reason, CapRAG uses CLIP (optimal for English retrieval) and CapRAG_M uses mSigLIP (optimal for multilingual retrieval). These are indeed the respective configurations that one would use if they wanted to do English-pivot retrieval or target-language retrieval and as such this is the most meaningful comparison we could provide here.
>
> > Why all unique tokens are considered valid concepts, when most tokens cannot be directly related to concepts?
>
> It is true that not all tokens in the concept lists are semantically meaningful or valid concepts. However, curating fully clean concept lists for all languages is a non-trivial task, especially for low-resource languages where reliable tools for filtering or validation are limited or unavailable. As a result, we chose to construct broad wordlists that include both valid and potentially noisy entries, with the goal of covering as many candidate concepts as possible. The core idea is that the retriever is responsible for selecting relevant entries from this noisy set. Rather than relying on pre-filtering, we let the retrieval process handle relevance, trusting that the learned similarity space will naturally favor semantically aligned and visually grounded tokens.
>
> > From the concept list, in the evaluation experiments, which concepts have been more often retrived and how do they relate to the images (this evaluation could be feasibly be done for the XM100 dataset)?
>
> We thank the reviewer for this insightful question. We previously attempted to evaluate the quality of retrieved concepts using precision and recall (using the retrieved concepts and the reference captions), but these metrics consistently produced low values, making it difficult to draw reliable conclusions. Instead, we analyzed the impact of concept list size in Figure 2, where we observed that expanding the wordlist on XM100 led to a drop in performance. This suggests that larger wordlists may introduce more irrelevant or unhelpful concepts, which are either not grounded in the image or not beneficial for the model during generation.
>
> > It would be interesting for future work to validate or order concept candidates by importance - e.g. through notions such as lexical salience [1] (often related to surprisal or predictability).
>
> We thank the reviewer for the suggestion. We did not conduct experiments on the ordering of retrieved concepts, as prior work (e.g., Wenyan et al., 2024 [1] ) has shown that retrieval-augmented image captioning models tend to be robust to the order of retrieved captions. However, we agree that it would be interesting to investigate whether this robustness extends to concept-level retrieval. We consider this a valuable direction for future work.
>
>
> [1] Li, W., Li, J., Ramos, R., Tang, R., & Elliott, D. (2024). Understanding Retrieval Robustness for Retrieval-Augmented Image Captioning. (https://arxiv.org/pdf/2406.02265)

---

> > ### Comment · Reviewer_H9Ar · 2025-06-06
> >
> > The authors have sufficiently replied to my questions.
> > I do not intend to change my score, as I think it adequately reflects the quality of the work

---

### Official Review · Reviewer_HrUv · 2025-05-12

**Rating:** 7
**Confidence:** 3
**Ethics Flag:** 1

**Summary:**

The paper proposes to improve the performance of multilingual image captioning by conditioning the generation on retrieved captions as well as retrieved image concepts (single words), and applies this idea to the mBLIP architecture.
The comparison with several baselines show improved performance on low and medium resourced languages. Ablating either caption or concept retrieval components shows that they both contribute to the preformance of the modified mBLIP.

**Reasons To Accept:**

The proposed methods is very simple while at the same time seems to improve captioning performance on the majority of languages.

**Reasons To Reject:**

The contibution is relatively minor modification of existing methods. The approach relies on retrieving the captions and concepts associated with related images, and thus may lead the model to rely more on generic (linguistic) priors rather than the specific content of the image itself. Given this risk, I would have liked an evaluation which focuses on ensuring that the model uses the visual input and does not "hallucinate".
---
Updated: The authors have addressed the concern about "hallucination" to some extent, and thus I'm upgrading my rating.

---

> ### Author Response · Authors · 2025-06-01
>
> We appreciate the reviewer’s comment. It is true that retrieval-augmented generation (RAG) introduces a linguistic prior, which in our case serves as a helpful complement rather than a replacement for visual grounding. To validate this, we included NoRAG (image-only) and CONCAP (image + text) in the paper, and are currently running Text-only ablations. In the Text-only setting, we replace the image with a solid-color version based on its mean RGB value, effectively removing visual content. This helps isolate the model’s reliance on retrieved text. Preliminary results, summarized in the table below, show that CONCAP consistently outperforms both NoRAG and Text-only across the 4 core languages, suggesting that the model benefits from both modalities and does not overly rely on retrieved text. Additionally, we are actively exploring methods for detecting hallucinations in a multilingual setting and are aiming to obtain the results as soon as possible.
>
> | Language | NoRAG | CONCAP | Text-only |
> |----------|-------|--------|------------|
> | English | 66.0 | 72.4 | 29.1 |
> | Spanish | 48.9 | 58.6 | 9.6 |
> | Chinese | 17.4 | 21.7 | 4.11 |
> | Hindi | 20.4 | 24.4 | 10.8 |
> | Avg_4 | 38.2 | 44.3 | 13.4 |

---

> > ### Author Response · Authors · 2025-06-06
> >
> > Dear Reviewer, we hope the above response adequately addresses your concerns. We kindly ask you to consider updating your score accordingly. We are happy to discuss your concerns further if needed.

---

> > > ### Comment · Reviewer_HrUv · 2025-06-10
> > >
> > > Thanks, increased my rating.

---

### Official Review · Reviewer_umB5 · 2025-05-13

**Rating:** 8
**Confidence:** 3
**Ethics Flag:** 1

**Summary:**

The paper introduces a multilingual image captioning model called CONCAP. Prior approaches faced a couple of issues including lack of large scale data and bias in translated captions. CONCAP improves the captioning for non-English languages by using retrieved captioned and image-specific concepts to augment caption generation. This approach outperforms state-of-the-art models on XM3600 dataset despite requiring much less training data. The performance is especially good on low and mid resource langauges.

**Reasons To Accept:**

- Substantial improvement on XM3600 while being highly data efficient.
- Novel approach for contextualization and grounding of captions using image-specific concepts helps address gaps in multilingual captioning performance.

**Reasons To Reject:**

- CONCAP fails to perform better than other approaches on certain high resource languages like en, es etc. The positioning of the paper could potentially be different to focus on mid and low resource languages since the approach is more effective for that setting.
- Retrieval is dependent on having a good English pivot. This could be a limitation if high quality translations are not available in certain scenarios.

---

> ### Author Response · Authors · 2025-06-01
>
> 1) As discussed in Section 4.8, CONCAP is particularly effective in low- and mid-resource language settings, where the model benefits more from the retrieved captions and concepts. In these cases, retrieval plays a more critical role in compensating for limited training data, which explains the stronger relative performance compared to high-resource languages like English or Spanish. We will emphasize this aspect more clearly in the camera-ready version to better reflect the strengths and positioning of our approach.
>
> 2) A key contribution of our work is identifying the reliance on English pivots in prior methods as a significant limitation. In real-world scenarios, this can become a limitation in scenarios where high-quality translations or English identifiers are unavailable. Our work specifically addresses this issue by demonstrating that concept-level retrieval in the target language remains effective even without English pivots, offering a more robust alternative for multilingual settings with limited or no English alignment.
>
>
>
> We thank the reviewer for the feedback and will ensure these points are more clearly emphasized in the camera-ready version.

---

> ### Comment · Reviewer_umB5 · 2025-06-09
>
> Thank you for the response. I do not wish to change my rating.

---

### Official Review · Reviewer_5aBu · 2025-05-13

**Rating:** 6
**Confidence:** 3
**Ethics Flag:** 1

**Summary:**

The paper proposes a multilingual image-captioning framework that augments an mBLIP backbone with two retrieval channels: (i) top-k captions from visually similar images and (ii) top-m lexical “concepts” retrieved from language-specific word lists. Both signals are concatenated into a prompt fed to the frozen mT0-XL decoder fine-tuned via LoRA. Training uses only 566k machine-translated COCO-35L pairs, yet CONCAP attains the highest average CIDEr on the 36-language XM3600 benchmark, surpassing the 7B parameter model while keeping few number of trainable weights.

-----------------After rebuttal------------------
Apologies for replying late while in the hospital - thanks for the rebuttal, which addresses my concerns. I have decided to increase my score.

**Questions To Authors:**

See weaknesses.

**Reasons To Accept:**

1. The motivation is clear.
2. The ablation is comprehensive.
3. The method achieves desirable performance with few number of trainable weights.

**Reasons To Reject:**

1. The evaluation is limited.
- Only the image captioning experiment on a single dataset XM3600 is provided. No retrieval, VQA or cross-modal generation tasks are provided.

2. English-pivot retrieval unrealistic: CapRAG’s strong gains hinge on English IDs unavailable in most real-world corpora; per-language retrieval (CapRAG-M) collapses.

3. Inference cost unreported: retrieving 4 captions + 10 concepts per image from 35 language-specific FAISS indices introduces latency and large memory footprints.

4. Several important analyses are missing.
- How sensitive is the model to the number of retrieved captions or concepts (e.g., what happens when n or m is increased or reduced dramatically)?
-  Could multilingual caption retrieval performance be improved by combining embeddings from multiple retrievers (e.g., ensemble of mSigLIP and mCLIP)?
- How does CONCAP perform in true zero-resource settings (i.e., no training data for a given language, only retrieval)?

---

> ### Author Response · Authors · 2025-06-01
>
> We thank the reviewer for the feedback. Below, we address the main points raised:
>
> 1) Our work is situated within a line of research on lightweight retrieval-augmented generation (RAG) for image captioning, and we have adopted the standard evaluation protocols prevalent in this specific area. This enables a clear and direct comparison with prior methods that share similar objectives and constraints. We agree that extending the evaluation to include tasks such as VQA would offer additional insights into the generalizability of our method. Although time constraints prevent us from conducting these experiments during the rebuttal phase, we will explore them in the next phase of this work. Based on the nature of our approach, we expect that the improvements observed in image captioning will carry over to VQA and other generative vision-language tasks.
>
>
> 2) We believe there may have been a misunderstanding. While prior work blindly opts for English-pivot retrieval, we explicitly highlight this as a key limitation for the reasons outlined by the reviewer (Section 4.4). Our work is the first to emirically show that in the absence of English parallel data, which is often the case in real-world settings, caption-based retrieval collapses.  Meanwhile, concept-based retrieval in the target language is significantly more stable and thus offers a viable alternative when English pivots are not available. This is a central contribution of our work which we will better highlight in the final draft.
>
>
> 3) Our work is positioned in the context of prior methods which rely on caption retrieval, with the novelty here being the additional concept retrieval. As such, below we discuss the increase in latency and memory footprint of caption+concept RAG compared to caption RAG.
> Table 6  in the appendix presents the wordlist length for each language, averaging approximately 43,940 words. MS-COCO contains 566,000 captions. This means that the memory footprint of the concepts index amounts to less than 10% of that of the captions index. We present a figure showing the size of caption and concept indexes at this link: https://imgur.com/a/4XLfzXV. Since concepts and captions can be retrieved in parallel, with retrieval from the newly added smaller index being faster by default, there is no added latency in the retrieval process.
> We plan to include a more detailed version of this analysis in the final draft.
>
> 4)
> - The sensitivity of the model to the number of retrieved concepts is explored in Table 3, where we experiment with retrieving 4, 10, and 20 concepts. The results show that performance is stable across these settings, with 10 and 20 concepts yielding similar scores (60.57 vs 60.19), and 10 slightly outperforming the rest.
> Since we follow the same caption retrieval setup as SmallCap [1], we adopt the optimal value for the number of retrieved captions, as obtained by Ramos et al. 2023 through hyperparameter tuning.
>
> - In principle, combining embeddings from multiple retrievers, such as mSigLIP and mCLIP, could improve multilingual caption retrieval. However, this comes with the practical drawback of increased inference cost due to the need to load and run multiple models simultaneously, and an increased memory footprint from storing multiple large indexes (each covering 566k captions). Moreover, it requires a study of the best way to combine the retrieved captions, possibly through learned functions, which is beyond the scope of our work.
>
> - This setting was explored in PAELLA [2], which showed that training on monolingual data can still yield strong zero-shot performance. Since ConCap follows a similar retrieval-augmented setup, we expect comparable zero-shot generalization. However, we do not explicitly target zero-shot image captioning in this work.
>
>
> [1] https://arxiv.org/abs/2209.15323
>
> [2] https://aclanthology.org/2024.findings-naacl.225/

---

> > ### Author Response · Authors · 2025-06-09
> >
> > Dear Reviewer, we hope our response adequately addresses your concerns. We kindly ask you to consider updating your score accordingly. We are happy to discuss your concerns further if needed.

---

> > ### Comment · Reviewer_5aBu · 2025-06-10
> >
> > Apologies for replying late while in the hospital - thanks for the rebuttal, which addresses my concerns. I have decided to increase my score.

---

> ### Author Response · Authors · 2025-06-06
>
> Dear Reviewer, we hope the below response adequately addresses your concerns. We kindly ask you to consider updating your score accordingly. We are happy to discuss your concerns further if needed.

---

### Decision · Program_Chairs · 2025-07-08

**Decision:**

Accept

**Comment:**

This paper proposes to augment the context of multilingual image captioning models with relevant retrieved image concepts in addition to retrieved captions. By doing so, the authors argue that it can counteract noise and improve the quality of generated captions. The proposed technique has been empirically validated through extensive experiments and ablation studies.

## Identified weaknesses
- The contribution of the paper is relatively incremental but reviewers acknowledge that while the approach is simple, it empirically shows that it is effective.
- The technique might over-rely on linguistic priors and ignore visual features, which could be a limitation in some cases. The authors addressed that issue by running additional ablation experiments which seem to indicate the models still rely on visual features to work.

## Positive aspects

- This work has clear motivation, provides thorough empirical evidence, and is well-structured.
- The authors show that their approach can surpass the SOTA model while training only a fraction of the parameters (111M vs. 7B) and with 10x fewer training data.
- This work identifies the dependence on English pivots in prior methods as a key limitation, particularly when high-quality translations or English identifiers are unavailable. It demonstrates that concept-level retrieval in the target language remains effective without English pivots, offering a more robust solution for multilingual contexts.

## Recommendation
Overall, the paper is a solid contribution to the field of multilingual image captioning. While the contribution may not be groundbreaking, it provides valuable insights and empirical evidence that can benefit both application and future research. The paper is recommended for acceptance, with the understanding that it is an incremental but meaningful advancement in the area of retrieval-augmented captioning. The authors have also made efforts to address reviewer concerns through additional experiments and clarifications.